# Early Succession of Community Structures and Biotic Interactions of Gut Microbes in *Eriocheir sinensis* Megalopa after Desalination

**DOI:** 10.3390/microorganisms12030560

**Published:** 2024-03-11

**Authors:** Wenlei Xue, Hao Wu, Xinyu Wu, Nannan Li, Ximei Nie, Tianheng Gao

**Affiliations:** 1Jiangsu Province Engineering Research Center for Marine Bio-Resources Sustainable Utilization, College of Oceanography, Hohai University, Nanjing 210024, China; 211311040027@hhu.edu.cn (W.X.); 211311040023@hhu.edu.cn (H.W.); 221311040020@hhu.edu.cn (X.W.); nanwl@hhu.edu.cn (N.L.); 2State Key Laboratory of Lake Science and Environment, Nanjing Institute of Geography and Limnology, Chinese Academy of Sciences, Nanjing 210008, China; 3Jiangsu Province Engineering Research Center for Aquatic Animals Breeding and Green Efficient Aquacultural Technology, College of Marine Science and Engineering, Nanjing Normal University, Nanjing 210023, China; nxm0529@163.com

**Keywords:** *Eriocheir sinensis* megalpopa, gut microbiota, larval development, bacterial community, gut co-occurrence network

## Abstract

As an enduring Chinese freshwater aquaculture product, the *Eriocheir sinensis* has a high economic value and is characterized by a catadromous life style that undergoes seawater–freshwater migration. However, little is known about their gut microbial status as they move from saltwater to freshwater acclimatization. Here, we sampled and cultivated *Eriocheir sinensis* megalopa from three aquaculture desalination ponds and investigated their gut microbiota diversity, community structures and biotic interactions from megalopa stage to the first juvenile stage after desalination for 9 days. Our results revealed that during the transition from megalopa to the first juvenile in *Eriocheir sinensis*, a significant change in gut microbial composition was observed (for instance, changes in relative abundance of dominant phyla), which was, however, not influenced by different sampling sites. The species diversity (such as the richness) of the gut microbiota showed a hump-shaped pattern along the succession. However, the compositional differences of the gut microbes showed constantly increasing patterns during the succession after freshwater adaption for all three sampling sites. Further co-occurrence analysis also showed that the complexity of the ecological networks in gut microbes was significantly enhanced during the development, such as increasing numbers of network links, connectivity and modularity, and was confirmed by decreasing average path length and proportions of negative links. Taken together, the differences in community structures and biological interactions of gut microorganisms were more pronounced in *Eriocheir sinensis* megalopa during desalination than in diversity and species compositions. This implies that the gut microbes of *Eriocheir sinensis* megalopa would become more robust and adaptive during the developmental process.

## 1. Introduction

The *Eriocheir sinensis*, a commercially valuable species, exhibits a descending oviposition cycle and migrates between freshwater and seawater [1,2]. Its life cycle typically covers a period of two years, which includes five life stages: egg, zoea, megalopa, juvenile and adult [3]. In brief, the *Eriocheir sinensis* survives in seawater during fertilized egg and zoea stages, then transitions to terrestrial rivers during the megalopa stage, gradually adapting to freshwater habitats [4,5]. After seawater desalination, *Eriocheir sinensis* megalopa are usually transported to hatcheries for further cultivation until the juvenile stage [6]. This practice of moving larvae from natural to controlled environments offers a unique opportunity to study the impact of environmental changes on the development and health of *Eriocheir sinensis* [7], particularly in understanding how these changes affect the gut microbiota during critical developmental stages.

Among the five life stages, the developmental stage from *Eriocheir sinensis* megalopa to the first juvenile larval stage is relatively special, as larvae at this stage need to undergo the acclimatization process that occurs during desalination. In addition, megalopa of *Eriocheir sinensis* shrink their abdomens and fold them under their chests to complete the metamorphosis from planktonic to benthic [8]. This morphological transformation is a critical period in the life cycle of *Eriocheir sinensis*, marking a significant transition in habitat and lifestyle [9]. Understanding the changes in gut microbiota during this period could provide insights into how these organisms adapt to new environments and the role of microbiota in their development and survival [10,11]. However, despite the significant habitat and developmental morphological changes during this particular stage, it has received little attention. This oversight leaves a gap in our understanding of how these changes influence the gut microbiota and, by extension, the health and development of *Eriocheir sinensis*.

Gut microbiota refers to the diverse group of microorganisms in the gut of animals that significantly influence the host’s growth, metabolism, and development [12,13]. Animal microbial communities shape after hatching or birth, improving gut, oral, and pores and skin microbial communities through continuous improvement. The microbial community in the early stages of life influences the composition of microbial communities throughout the life cycle, with long-term implications for host health [14,15,16]. More and more research is focusing on understanding the process of microbial community succession in aquatic animal larvae, such as *Litopenaeus vannamei* [17] and fish [18]. These studies have shown that early microbial colonization can have lasting effects on the host, affecting everything from disease resistance to nutrient absorption. However, the current information on the composition and succession of microbiota in *Eriocheir sinensis* megalopa during developmental stages is not available.

Biological migration is usually associated with disturbances and environmental changes that are key drivers in determining the diversity, community compositions, and function of the gut microbiome [19]. Most of the research of the intestinal microbiota of *Eriocheir sinensis* has centered on their composition and dynamics in response to other factors [20,21,22], neglecting the study of the succession of gut microbiota during the larval stage. Previous studies on crustacean larvae have shown that the bacterial community composition of juvenile shrimp plays an important role throughout the life cycle, and the microbiome of juvenile shrimp can be considered a trading point between the post-larval and adult stages [18,23]. Some studies indicated significant differences between planktonic and animal gut bacterial communities in aquatic environments [24,25]. A recent study found that *Litopenaeus vannamei* contain only a small amount of gut microbiota from their aquatic environment [26]. On the other hand, a survey of *Gadus morhua* indicated that larvae had been strongly affected through the means of the water bacterioplankton [27]. These inconsistent effects suggested that species specificity may also have an effect on gut microbiota outcomes. Therefore, it is essential to understand the succession of the microbial community in the *Eriocheir sinensis* megalopa.

In this study, we utilized the habit of *Eriocheir sinensis* megalopa undergoing changes in habitat and developmental morphology after the freshwater adaptation stage to investigate how host development and environmental adaptation affect the assembly and succession of intestine microbiota. To achieve this, we collected samples of *Eriocheir sinensis* megalopa (adapted to freshwater conditions) from three different spawning grounds (Dongtai, Rudong, and Sheyang) and transferred them to the laboratory for unified feeding. The experimental water bodies were all drinking water. The purpose was to remove the influence of other environmental factors besides the water environment, and after a period of development, the megalopa of three different spawning grounds would eventually develop into a juvenile crab. We have three goals: (1) to ascertain whether the established gut microbiota of *Eriocheir sinensis* megalopa will be influenced by switching into different environments, (2) the function of the succession of community structures and diversity of gut microbiota in *Eriocheir sinensis* megalopa during the developmental times, and (3) how will the interaction patterns of gut microbiota change towards the succession?

## 2. Materials and Methods

### 2.1. Experimental Design and Sample Collection

We first collected *Eriocheir sinensis* megalopa from three aquaculture desalination ponds in Dongtai County (32.883° N, 120.968° E), Rudong County (32.218° N, 121.381° E) and Sheyang County (33.895° N, 120.449° E) in the eastern coastal region of China (Figure 1). Specifically, we collected *Eriocheir sinensis* megalopa from three aquaculture desalination ponds in Dongtai, Rudong, and Sheyang, and transported them to Hohai University for subsequent rearing, where all *Eriocheir sinensis* megalopa were uniformly reared in sterile plastic baskets with dimensions of 0.6 × 0.5 × 0.4 m^3^. In order to simulate the natural situation of breeding *Eriocheir sinensis* megalopa in the hatchery after seawater desalination, all samples were tested directly without acclimatizing to the environments. Specific measures, including continuous aeration and control of water conditions, continuous feeding of hydrilla (the most common feed to ensure basic survival of megalopa) and maintenance of pH (8.1 ± 0.1), dissolved oxygen (8.5 ± 0.2 mg/L), salinity (0 PSU) and ammonia nitrogen levels (NH_4_^+^-N < 0.1 mg/L) at normal levels were carried out to ensure the stability of the culture environment. We measured salinity, DO, and pH using the multiparameter probe YSI 556 MP, and the concentration of ammonia nitrogen (NH_4_^+^-N) was analyzed and confirmed using Nessler’s reagent spectrophotometry according to the Chinese Ministry of Ecology and Environment’s standards for water quality analysis [28]. For megalopa samples, surfaces were rinsed using sterile water and hindgut contents were collected by dissecting needles. According to the different developmental stages of *Eriocheir sinensis* after the completion of desalination [19], we collected intestinal samples a total of three times, the first time in Dongtai, Rudong and Sheyang to collect *Eriocheir sinensis* megalopa in desalination ponds regarded as the initial stage of the completion of desalination (Day 1), and the second and third time were both in the Hohai University sterile plastic baskets. The second and third time were collected in sterile plastic baskets at Hohai University as the environmental adaptation stage (Day 5) and the developmental completion stage (Day 9), in which the collected samples were in the *Eriocheir sinensis* megalopa stage on Day 1 and Day 5, and in the first juvenile stage on Day 9, so a total of 54 Chinese mitten crab intestinal samples were collected (that is, 3 time points × 3 sites × 6 replicates). The samples were next stored in sterile tubes at −80 °C until further molecular analysis.

### 2.2. DNA Extraction and MiSeq Sequencing

Bacterial 16S rDNA amplicon evaluation was carried out once on all intestinal samples (n = 54) through Illumina sequencing. Considering the small size of the individual *Eriocheir sinensis* at the larval stage and the inadequate quantity of DNA bought from the gut, we pooled the three intestines to form one biological replicate, and every six biological replicates to form one group. Total genomic DNA was extracted using the OMEGA kit E.Z.N. A™ Mag-Bind Soil DNA Kit (MoBio, Carlsbad, CA, USA), followed by the verification of DNA concentration using 0.8% agarose gel electrophoresis. The purity of genomic DNA was detected by 0.8% agarose gel electrophoresis and its concentration was determined with Qubit^®^ 3.0 fluorometer (Invitrogen, Waltham, MA, USA). The V3 and V4 highly variable regions of 16S rRNA were amplified using universal primers 341F (5′-CCTACGGGNGGCWGCAG-3′) and 805R (5′-GACTACHVGGGTATCTAATCC-3′), and PCR was performed in 30 μL of reaction: 20 ng of DNA template, 15 μL of 2× Hieff^®^Robust PCR Master Mix (Takara Biotechnology, Dalian Co., Ltd., Dalian, China), 9 to 12 μL of H_2_O, and 1 μL each of the primers. Amplification conditions were: denaturation at 94 °C for 3 min, followed by 5 cycles (94 °C, 30 s → 45 °C, 20 s → 65 °C, 30 s). Finally, 20 cycles were performed (94 °C, 20 s → 55 °C, 20 s → 72 °C, 30 s). After the cycles ended, the final extension was 72 °C for 5 min. The PCR amplicon size was detected using 2% agarose gel electrophoresis, and the library concentration was decided using the Qubit 4.0 fluorometer (Invitrogen, USA). The amplicons from every reaction mixture were blended in an equimolar ratio based on their concentration. According to the manufacturer’s instructions, a Illumina MiSeq system 2 × 300 bp platform (Illumina MiSeq, Carlsbad, CA, USA) was used for sequencing. Raw sequence reading were filtered and subjected to primer removal using DADA2 in QIIME2 V2022.8.3 [29,30,31]. These obtained filtered readings were then de-duplicated and subjected to denoising. After merging paired-end sequences and removing chimeric sequences, readings were filtered using a quality control score of >25. The taxonomic classification of each ASV of bacteria and fungi was obtained using the SILVA 138 and UNITE databases with the *q2-feature-classifier* (sequence identity = 100%, using USEARCH *closed_ref* command) [32,33]. The ASV tables of each group and dataset in USEARCH tabular format were imported into R (version 3.6.0, R Core Team, 2019) and merged into the integrated ASV tables for further analysis. ASVs assigned to ‘mitochondria’, ‘chloroplasts’, ‘archaea’, and ‘eukaryotes’ were removed from the bacterial dataset.

### 2.3. Statistical Analyses

The results were presented in a box plot after using QIIME2 V2022.8.3 to calculate alpha diversity (ACE index, Chao1 richness index, Richness index, Shannon diversity index, Pielou’s evenness, Simpson index, Invsimpson index, observed species richness and Good’s coverage). Principal coordinate analysis (PCoA) was used to visualize the Bray–Curtis distance matrices used to calculate beta diversity. The Bray–Curtis distance carried out in the QIIME2 V2022.8.3 program was used to detect differences between groups by conducting an analysis of similarities (ANOSIM). To investigate the effects of different sampling locations and times, we analyzed changes in the composition (here representing the levels of Phylum and Family of the microbial communities) as well as the diversity of the gut microbial community using mixed-effects modeling. Specifically, for different Phylum (or Family and diversity), we performed the model: y ~ Time × Site + (1|Rep) to examine the influences of sampling site and time. Next, we conducted post hoc tests (the Type III analysis with Satterthwaite’s method) to determine if differences across sampling time and sites were significant, and sorted them alphabetically. Correlation analysis of SparCC from the “Speci-easi” R package was employed to assemble the association network to depict the complex coexistence patterns in gut microbiome networks. The SparCC is an algorithm that considers sparse component data in network construction, and is more combinatorial robust than the traditional correlation analysis [34]. To ensure the reliability of the correlation calculations, species that occurred in at least 20% of the samples were selected [35]. We used 100 permutations to calculate the significance of correlations between taxon abundances and retained correlation matrices with correlations > 0.4 and significance < 0.05 for subsequent network construction. We included various network topological parameters to represent the complexity of species interactions among microbes [36,37], including: (1) Nodes: number of microbial taxa in the network. (2) Links: number of connections between each pair of taxa. (3) Connectivity: ratio between the sum of the actual number of connections between taxa in a network to the total number of potential connections. (4) Average path length: average distance of all node pairs in the network. (5) Modularity: the degree to which the network is divided into different sub-communities or modules. (6) Proportion of negative links: ratio of negatively correlated links to all links. All the parameters were calculated using the package “igraph” in R. [38,39]. To determine whether the differences between comparisons are significant, significance tests were conducted using the Wilcoxon test.

## 3. Results

### 3.1. Analysis of the Difference in the Intestinal Microbial Community

To discover the variations in community composition and variety over developmental time in different locations, we sequenced 54 samples from the three spawning grounds and obtained a total of 4,027,465 sequences, identifying 3402 different OTUs which represent 852 genera.

The Venn diagrams (Appendix A) were used to examine the distribution of OTU in different groups. The Day 9 group recorded the highest number of OTUs (2003), with Day 1 having the second highest number (1807), and Day 5 having the lowest (1692). According to the Venn diagram, all groups shared 768 core OTUs, and exclusive OTU ratios were highest on Day 9 (61.7%) and Day 1 (55.5%). These findings demonstrated that developmental time can lead to more complex microbial diversity.

We found that the succession of *Eriocheir sinensis* strongly affected the composition of gut microbes more than the sampling sites. The dominant Phyla among the gut microbial community are *Proteobacteria* (67–72%), *Actinobacteria* (11–15%), and *Euryarchaeota* (7–15%; Figure 2A), and the dominant Genus are *Rhizobium* (27–38%), *Bosea* (27–34%) and *Acinetobacter* (3–12%; Figure 2B). Notably, both of the phylum and genus levels contained “Unassigned” and “Others”, where the former represents the species that mismatched with all species responding to the database (SILVA 138) used for microbial identity, and the latter represents the species that exist in a level that is not a part of all others, respectively. Further analysis showed that the sampling time had a greater effect on gut microbial community structure than sampling location (Table 1). Specifically, 60% of the dominant Phylum and 70% of the dominant Genus were significantly affected by changes in sampling time, while only 20% of the Phylum and Genus were significantly changed with the sampling sites. In addition, the interactive effects between the sampling time and sites were greater at the Genus level (40%) than at the Phylum level (20%). Taken together, our results emphasize that the gut microbial composition of *Eriocheir sinensis*, although relatively similar between three sampling sites, would change significantly over succession for most dominant phylum (or genus).

### 3.2. Diversity of Bacterial Communities

To demonstrate the structural complexity of each sample, we first calculated microbial alpha diversity using various metrics. The results showed that alpha diversity measured by the Shannon, Simpson, Chao1 and ACE indexes and Pielou’s evenness was the highest for the Day 5 group, although it was not significantly different from that of other groups. All samples had a hump-shaped trend in their diversity and richness (Figure 3). However, the diversity index analyses across all samples revealed no significant differences in the developmental stages of *Eriocheir sinensis* megalopa. The Good’s coverage of every sample, which determines the sequencing completeness, was >99.9%, suggesting that the identified sequences symbolize the majority of the bacteria in every sample.

Next, PCoA based on Bray–Curtis similarity distance was applied to visualize the difference in community of *Eriocheir sinensis* megalopa intestines at the three different spawning sites and three different developmental periods (Figure 4A,B). The PCoA analysis showed that the distribution structure of gut bacterial communities in the three different spawning grounds was spatially closer and there was no significant difference. Interestingly, from a timescale perspective, the gut bacterial community of *Eriocheir sinensis* megalopa at different developmental periods becomes more dispersed over time, with dense aggregation on Day 1 and Day 5, whereas it is more dispersed on Day 9. This may be because the early stages of freshwater adaptation are more susceptible to environmental changes, while individuals in the developmental stage become more unique, resulting in a more dispersed distribution.

To further investigate the succession of gut microbiota in three different spawning grounds during the development process of the *Eriocheir sinensis* megalopa, we calculated and standardized the differences in the communities of different samples using the Unifrac index and the Sorensen index, respectively, and compared the samples at different developmental times based on the Wilcox’s *t*-test, and obtained the differences in the communities on Day 5, Day 9 and the initial community, respectively. The consequences indicated that the freshwater adaptation of three spawning grounds significantly improved the log10-transformed of Unifrac distance in the later stage compared to the early stage (Figure 4C). Interestingly, the log10-transformed of Sorense distance showed that the DT (Dongtai) group improvement significantly compared to the early stage, with no significant change in the RD (Rudong) and SY (Sheyang) group (Figure 4D).

### 3.3. Gut Microbial Co-Occurrence Networks

To comprehend the significance of biological interaction in the formation of communities, symbiotic networks of intestinal microbiome (OTU level) of *Eriocheir sinensis* megalopa were constructed based on the significant correlation among three groups of different sites in DT (Dongtai), RD (Rudong) and SY (Sheyang) (Figure 5A). The consequences showed that in the DT group, the levels of edge and node connectivity are higher, while the average path length values are lower, indicating that the interactions between bacteria communities are becoming more significant and complex. Most of the network topology characteristics of the *Eriocheir sinensis* megalopa gut microbiome showed significant differences during development (*p* < 0.05) (Figure 5B). During development, the negative correlation ratio and average path length show a decreasing trend, while connectivity, nodes, links and modularity increase over time (Figure 5B).

## 4. Discussion

Recent studies have explored the critical roles that the gut microbiomes play in host food digestion, development, homeostasis and immune protection [40,41]. Conversely, the gut microbiota is influenced by several factors, such as genetics, developmental stage, habitat environment, feed type and health status [42,43]. As a migratory species, the *Eriocheir sinensis* usually undergoes habitat and developmental morphological changes during the megalopa stage, but little is known. In this study, by using Miseq sequencing technology, three different spawning grounds of megalopa were characterized in terms of their microbial composition and diversity succession. In addition, symbiotic analysis was used to study the interactions between microorganisms in the gut microbial network. Our findings indicated that the gut species composition did not change significantly during succession, while the microbial diversity exhibited a hump-shaped pattern. The structure and network complexity of gut microbial communities increased significantly during succession.

First, we found that the relative abundance of the gut microbiota composition of *Eriocheir sinensis* megalopa (especially the core species) changed significantly with freshwater acclimatization, while a weaker correlation was observed with the sampling site (Figure 3, Table 1). Here, the dominant species of the gut microbiota belonged to the phylum of *Proteobacteria*, *Actinobacteria*, and *Bacteroidetes*, which is similar to the results of previous studies. For example, Zhu et al. found that these three phyla were dominant in gut of adult *Eriocheir sinensis* [44]. Bacteria from different phyla usually show differences in maintaining important intestinal functions and homeostasis, thus changes in their relative abundance may not only indicate changes in the external environment, but also characterize differences in host gene expression as well as developmental processes [45,46,47]. For example, *Actinobacteria*, as a group of Gram-negative bacteria, play essential functions in the host immune system, protecting daily immune characteristics [48]. In addition, typical gut microbial dominant phyla such as the *Firmicutes* phyla consist of a large number of more functionally diverse core bacteria, whose changes in relative abundance often represent alterations in tolerance and are associated with protein metabolism and host body weight [49,50,51]. Taken together, our results suggest that the effect of developmental time may be much greater than the difference in distance in the process of adaptation to desalination in *Eriocheir sinensis* megalopa.

Differences in the structuring of microbial communities at different developmental times may further lead to changes in the proportions of endemic and shared species (Appendix A). We showed that the proportion of endemic species to overall species of gut microbes was much higher in the pre-desalination period (38.7% on Day 1) and the post-desalination period (41.6%) than in the intermediate period (31.2%). This result implies that the short-term community structure of gut microorganisms adapted to the environment undergoes dramatic fluctuations and is most unstable during the plateau period. This is similar to that of Shao et al., suggesting that the species richness of the gut microbiome varied in a hump-shaped manner over time during the seawater–freshwater transition phase, and that salinity was the main factor contributing to the variability of the gut microbiome community in *Eriocheir sinensis* [19]. In addition, an increase in the proportion of endemic species with the process of acclimatization may indicate an increase in the homeostasis of the intestinal environment and may be progressively more different from the early stages of development as well as from the surrounding environment [52,53]. For example, the gut microbiota of the vertebrate zebrafish (*Danio rerio*) tends to aggregate into distinct communities throughout development, and these communities become increasingly different from their surroundings and from each other [54]. Microbiota may be particularly important at key developmental stages, as found in wood frogs, where disruption of the microbiota during early larval life was found to have a legacy effect on development that persisted until later, long after the microbiome had recovered from the perturbation [55]. The composition of a species’ gut microbial community can change seasonally and as the host moves between environments (e.g., freshwater and saltwater for migratory fish) [56,57]. It can also change with the host’s dietary choices, metabolic rate (or needs), and life stage [58,59]. These changes in the composition of the gut microbial community allow the host to live in different environments, adapt to seasonal changes in diet, and maintain performance throughout the life history, highlighting the ecological relevance of the gut microbiota [60].

The alpha diversity of bacteria in the host-associated microbe is known to promote its stability and function and increase its diversity can also improve the health of the host [61,62]. Our results showed that the alpha diversity of microbes showed hump patterns in guts during developments (Figure 4). This finding is consistent with observations in *Eriocheir sinensis* and other animals such as Atlantic salmon [19,63]. During the process of the seawater–freshwater transition, the species richness of the gut microbiome of *Eriocheir sinensis* shows a hump-shaped trend, while the abundance of the gut microbiome of Atlantic salmon shows a hump-shaped pattern after transferring from the same hatchery to circulating aquariums and open lakes. We speculate that due to the incomplete development of the larvae, the larval microbial community is mainly derived from the external environment and internal organs. With increasing development time, the “internal” gut microbiota of the larvae begins to develop. With changes in the external environment and completion of self-development, the internal and external gut microbiota will undergo recombination, leading to changes in the gut microbiota. Generally speaking, high alpha diversity means that the microbial community in the host physique has greater stability, while low alpha diversity means that the microbial community in the host physique is extra fragile [64]. After freshwater adaptation (Day 1–Day 5), the alpha diversity of bacteria in the megalopa increases, which may be due to environmental changes. In order to adapt to external pressure, the *Eriocheir sinensis* megalopa require high and stable alpha diversity to resist external pressure. During the developmental stage (Day 5–Day 9), the diversity of the bacterial alpha in megalopa decreases, which may be due to the fragility of the megalopa caused by molting. These findings may be a key reason for the high mortality of *Eriocheir sinensis* megalopa during the molt to the first stage of juvenile crabs.

The core microbiota, which can be found in the intestine, are long-term stable colonies. They play a vital role in nutrient absorption, immune response, and physiological metabolism [65]. The Venn plot results show that all groups share 768 core OTUs, with the highest number of exclusive OTUs on Day 9, suggesting that developmental time may lead to more complex microbial diversity. Interestingly, the results of the Lefse analysis also support this point, with more bacteria significantly enriched in the gut of the Day 9 group. Compare the similarity of microbial populations between different developmental stages of *Eriocheir sinensis* megalopa by PCoA. The results suggest that host development may lead to a more dispersed gut bacterial community. This result was steady with previous research, which reported that development could lead to a more mature and complex host gut microbiota, and the more mature the organism, the greater the differences in individual microbial community structure [54,66,67]. Therefore, we concluded that the host was uncovered to microbial successions during ontogeny. It is viable that this supports the thought that morphological development leads to adjustments in the microbiota and vice versa. According to this concept, the gradual adaptation of the host microbiota, leading to changes in the main gut microbe, is at least partly explained by the different gut microbiota of *Eriocheir sinensis* at different developmental stages.

Network analysis could be used in the study of microbial interactions and the statistical identification of taxa that are highly connected in the community [68,69,70]. Microbial interactions are crucial for maintaining the stability and diversity of communities. We found that the microbial ecosystem networks in the first juvenile crab were more stable than in *Eriocheir sinensis* megalopa, and that the development of *Eriocheir sinensis* megalopa notably improved microbial interactions in the intestine ecosystem. Previous studies have shown that this is consistent with the analysis of gut microbiota networks in fish [71]. Our study shows that with the development of *Eriocheir sinensis* megalopa, the gut microbial ecosystem has a positive impact, which is manifested as the gut microbe in the developmental stage having greater nodes, links, connectivity, and modularity compared to the early stage, which often means that the network is more complex and tightly connected, thus improving the stability and system efficiency of the entire network [72]. The study also found that in the early stages of freshwater adaptation, the gut microbiota network of the *Eriocheir sinensis* megalopa showed a high degree of negative correlation, indicating a high degree of ecosystem stability. Generally, cooperating networks of microbes tend to be unstable, but microbial competition could enhance the stability; this means stability could be promoted by increasing negative correlations or limiting positive feedbacks [73]. It is widely known that modularity is a structural characteristic of networks, which means that they are divided into well-defined sub-modules [74]. Moreover, mean connectivity had a positive correlation with network complexity. Thus, the complexity of the ecological network was enhanced by host development, which suggests a positive impact on the gut microbial ecosystem [43,75]. To summarize, *Eriocheir sinensis* megalopa development led to an expansion in microbial interactions and balance of the intestine ecosystem. Future microbial management can benefit from these findings in providing higher intestine ecosystem offerings for the host.

## 5. Conclusions

In this study, we investigated the gut microbiota diversity, community structures and biotic interactions of *Eriocheir sinensis* from megalopa stage to first juvenile stage after freshwater adaptation. The results showed that during the development process from megalopa to first juvenile in *Eriocheir sinensis*, no significant change in gut microbial composition was observed; it was found that the diversity and composition of *Eriocheir sinensis* megalopa bacterial community dynamically varied with host development, with the hump-shaped pattern of alpha diversity. In addition, the shape of the intestine microbial community and the ecological network of the *Eriocheir sinensis* megalopa have become increasingly complex with the process of development. Altogether, our results can help understand the mechanism of gut microbiota succession in *Eriocheir sinensis* following freshwater adaptation and furnish a reasonable theoretical foundation for the health status of the *Eriocheir sinensis* during the megalopa stage.

To deepen our understanding of the changes in the gut microbiota of *Eriocheir sinensis* throughout the host’s lifecycle, future research should adopt the following strategies: First, it is essential to strengthen longitudinal studies on the diversity of gut microbiota in *Eriocheir sinensis* at different growth stages. By extending the research to include the host’s adulthood and breeding periods, we can comprehensively reveal the dynamic changes in the gut microbiome of *Eriocheir sinensis* throughout its lifecycle, providing a basis for uncovering how microbial communities respond to changes in the host’s physiological state [76,77]. Second, it is crucial to delve into the interaction between gut microbiota and host health. Specifically, future research should focus on how structural changes in the gut microbiota can impact the host’s physiological and health status, especially how gut microbiota facilitate the host’s growth and survival in adapting to new environments [53,78]. Additionally, utilizing high-throughput sequencing technologies, such as metagenomics, to study the functional characteristics and metabolic pathways of gut microbiota will help to further understand how they support the host in adapting to freshwater environments [18,79]. This approach will not only reveal changes in microbial composition but also deepen our understanding of how these changes impact the host’s adaptability and survival through specific metabolic pathways and functional characteristics. In summary, by considering research strategies that incorporate growth stages, host health, and microbial function, we will be provided with a comprehensive framework to understand how the gut microbiota of *Eriocheir sinensis* functions throughout the host’s lifecycle.

## Figures and Tables

**Figure 1 microorganisms-12-00560-f001:**
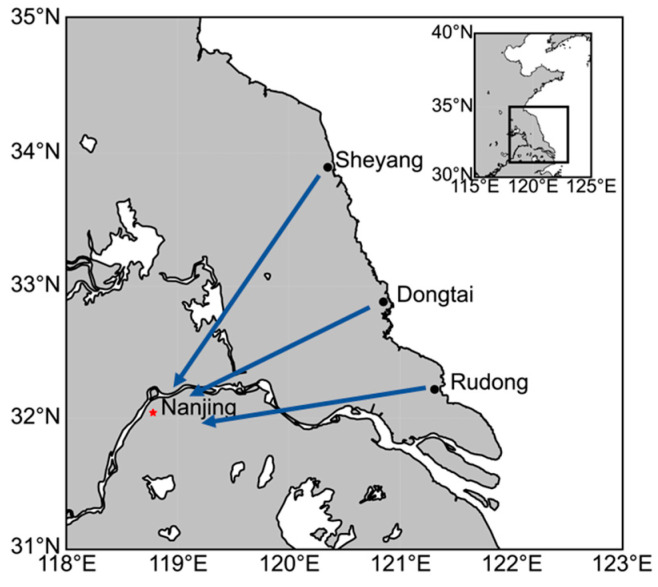
Sampling sites in this study.

**Figure 2 microorganisms-12-00560-f002:**
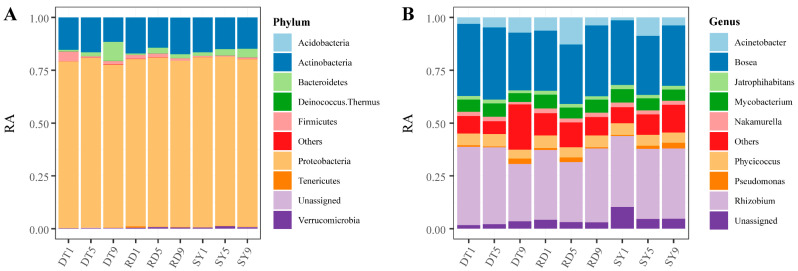
Relative abundance of intestinal microbes in *Eriocheir sinensis* megalopa at the Phylum level (**A**) and Genus level (**B**).

**Figure 3 microorganisms-12-00560-f003:**
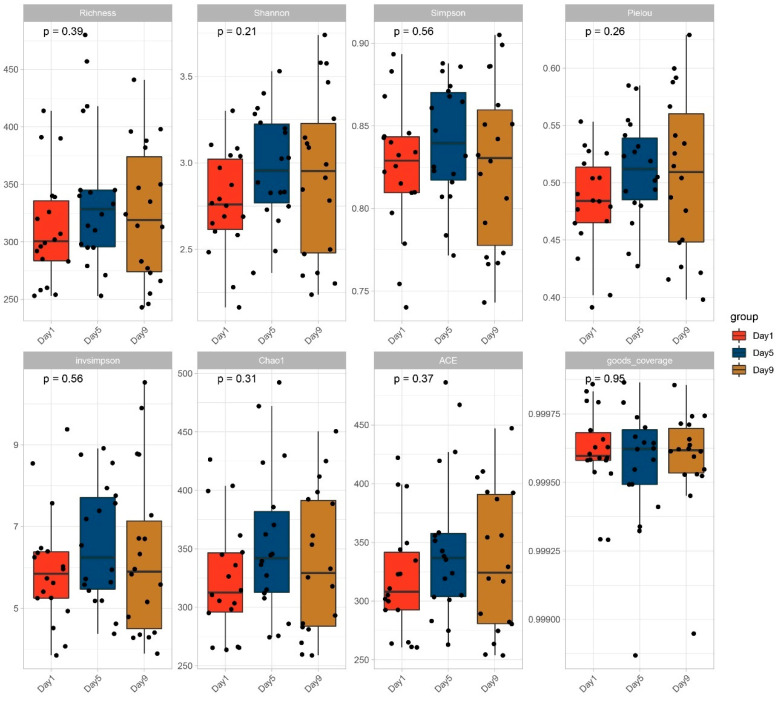
Differences in alpha diversity measures (Richness, Shannon index, Simpson index, Pielou’s evenness, invsimpson index, Chao1 index, ACE index and Good’s coverage) of the bacterial species in the intestines of megalopa at different developmental time.

**Figure 4 microorganisms-12-00560-f004:**
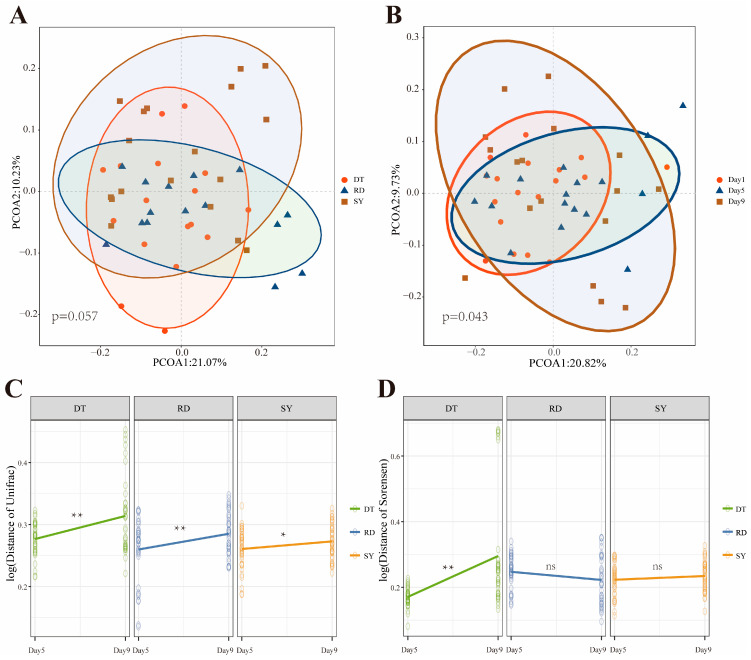
Differentiation and succession of gut microbiota in *Eriocheir sinensis* megalopa over development time. Principal coordinate analysis (PCoA) visualizing compositional variations of *Eriocheir sinensis* megalopa gut bacterial communities in different spawning sites based on Bray–Curtis dissimilarity (**A**) and different developmental times (**B**). Based on Wilcox’s *t*-test was conducted to compare and obtain the differences between the community and the initial community on the Day 5 and Day 9, respectively, visualizing and standardizing the differences between different sample communities using the Unifrac index (**C**) and the Sorensen index (**D**). The “ns” stands for no significant difference, * *p* < 0.05, ** *p* < 0.01 represent significant differences.

**Figure 5 microorganisms-12-00560-f005:**
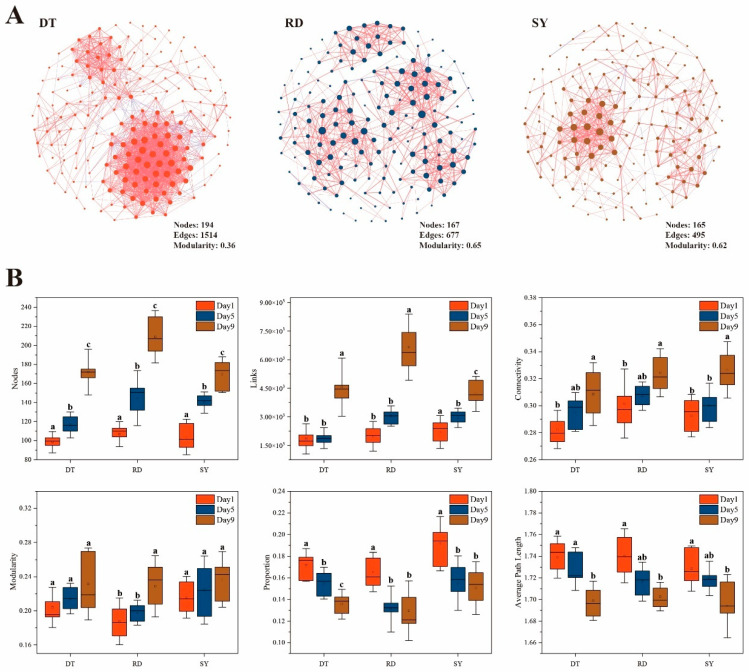
Total co-occurrence network of the gut microbial community. (**A**) The co-occurrence network of the gut microbiome of *Eriocheir sinensis* in DT, RD, and SY group at the OTU level. Each of the nodes represents unique OTUs in the data sets. The size of each node is proportional to the number of connections (that is, degree). OTUs are colored by different sites. The red side is positive correlation and the blue side is negative correlation. (**B**) The relationship between network topology characteristics and time of gut microbiome communities. Different lowercase letters indicate significant differences (*p* < 0.05) at different times in the same group. Bars represent the mean ± SD (N = 18).

**Table 1 microorganisms-12-00560-t001:** Effects of sampling sites and time on the gut microbial community compositions. Here, changes in the 10 dominant phyla were determined using the linear mixed-effects model of “y ~ Time × Site + (1|Rep)”, where the “Rep” represent the replication groups. Type III analysis with Satterthwaite’s method was used to determine if differences across sampling time and sites were significant. We show the results of F and significant *p* values in the table (* 0.01 < *p* ≤ 0.05; ** 0.001 < *p* ≤ 0.01; *** *p* ≤ 0.001).

Phylum	Site	Time	Site × Time
Proteobacteria	0.06	0.46	0.05
Actinobacteria	0.05	**3.12 ***	0.89
Bacteroidetes	0.31	**4.04 ***	1.20
Firmicutes	**4.88 ****	**3.43 ***	2.25
Verrucomicrobia	0.37	0.09	0.04
Tenericutes	**5.01 ****	1.44	**2.89 ***
Unassigned	0.40	**8.38 *****	**3.37 ***
Acidobacteria	0.27	1.46	1.31
Deinococcus.Thermus	0.30	**3.29 ***	0.59
Others	0.74	**2.89 ***	1.62

## Data Availability

The raw sequencing data can be found at the National Centre for Biotechnology Information (NCBI) Sequence Read Archive (SRA) with an accession number: PRJNA1058366.

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
