# Peer review of "Early Succession of Community Structures and Biotic Interactions of Gut Microbes in Eriocheir sinensis Megalopa after Desalination"

_microorganisms, 2024, doi:10.3390/microorganisms12030560_

Round 1
Reviewer 1 Report
Comments and Suggestions for Authors
Dear Editor,
In the study titled "Stability of Gut Microbial Composition During Developmental Stages of Eriocheir sinensis Megalopa," the authors explore the dynamics of gut microbiota in the Chinese freshwater aquaculture product Eriocheir sinensis megalopa during its transition from saltwater to freshwater habitats. The study aims to investigate the stability of gut microbial composition and diversity during this critical developmental stage.
Overall, the study provides valuable insights into the stability of gut microbial composition during the development of Eriocheir sinensis megalopa and underscores the importance of considering species-specific factors in microbial ecology research.
The work requires further clarification and additional explanation on the following aspects:
The authors should more carefully justify the observed stability of gut microbial composition and its implications for this species.
The authors should clarify and emphasise why, according to the results of the study, the gut microbial composition of Eriocheir sinensis megalopa remained relatively stable before and after adaptation to fresh water, in contrast to the results of other organisms in which the gut microbial composition was influenced by environmental factors and host development. It is not well understood.
The authors should also investigate why the observed stability of gut microbial composition during the development of Eriocheir sinensis megalopa may be related to the stability of the aquatic environment and the short developmental cycle of this species.
The authors should elucidate in more detail the potential influence of environmental factors and host developmental stages (have such studies been done previously and by whom?) on the dynamics of the gut microbiota.
Also important in this work are the detailed suggestions, which the authors have not fully developed, for future research directions to elucidate the mechanisms underlying the observed phenomena.
The authors should further elaborate on the broader implications of these findings for aquaculture practice and ecosystem management.
Additional clarity and detail in these areas will enhance the completeness and depth of the study, providing a clearer understanding of the dynamics of the gut microbiota in Eriocheir sinensis megalopa.
Importantly, the paper reads heavy, with many large parts in the text without structuring, which reduces the value of presentation and selection of results by the reader. The authors should improve the text to make it fully presentable.
In summary, it is evident that the authors need to address and clarify several key issues in their work to enhance its comprehensiveness and impact. Specifically, they should provide additional detail and explanation regarding the methodology employed for microbial characterization and symbiotic analysis. Furthermore, the rationale behind the observed stability in gut microbial composition and its implications should be elucidated. Additionally, considering the potential influence of environmental factors and host developmental stages on gut microbiota dynamics is crucial. Suggestions for future research directions would also be valuable in shedding light on the underlying mechanisms driving the observed phenomena. Lastly, improvements in the clarity, organization, and presentation style of the manuscript, including data presentation, citations, language, and engagement, are essential for ensuring the work's effectiveness and readability. Overall, addressing these issues and making necessary revisions will significantly strengthen the quality and impact of the study.
Author Response
Dear Reviewer,
Thank you very much for your comments and professional advice, these opinions help to improve academic rigor of our article. Please refer to the attachment for specific modification details. Thank you again for your attention and time. Looking forward to your reply.

Reviewer 2 Report
Comments and Suggestions for Authors
The work is interesting, but not explained clearly enough. The experimental part needs to be refined as well as the introduction.

Author Response
Dear Reviewer,
Thank you very much for your comments and professional advice, these opinions help to improve the academic rigor of our article. Please refer to the attachment for specific modification details. Thank you again for your attention and time. Looking forward to your reply.

Round 2
Reviewer 1 Report
Comments and Suggestions for Authors
Dear Editor!
Authors have addressed all the comments in the article, and it is now ready for submission for publication.
Author Response
We thank the reviewer for the positive remarks and for finding our manuscript ready for publication.
Reviewer 2 Report
Comments and Suggestions for Authors
The paper can be accepted in this form.
Comments on the Quality of English LanguageThe English is ok.
Author Response
We appreciate the comments provided by Reviewer 2 along the review process, which we believe have contributed to a stronger manuscript.